# Magnitude and associated factors of poor medication adherence among diabetic and hypertensive patients visiting public health facilities in Ethiopia during the COVID-19 pandemic

Tariku Shimels[1]*, Rodas Asrat Kassu[2], Gelila Bogale[3], Mahteme Bekele[1], Melsew Getnet[1], Abrham Getachew[1], Zewdneh Shewamene[4], Mebratu Abraha[1]

1 Research Directorate, Saint Paul's Hospital Millennium Medical College, Addis Ababa, Ethiopia,
2 Department of Neurology, Saint Paul's Hospital Millennium Medical College, Addis Ababa, Ethiopia,
3 United Vision Medical Services, Addis Ababa, Ethiopia, 4 Ethiopian Health Insurance Agency, Addis Ababa, Ethiopia

* tarphar2008@gmail.com

## Abstract

### Objective

This study aims to assess the magnitude and associated factors of poor medication adherence among diabetic and hypertensive patients visiting public health facilities in Addis Ababa, Ethiopia during the COVID-19 pandemic.

### Methods

A multi-site cross-sectional design was conducted from 1st through 30th of August 2020 at public health facilities of the study area. Adult outpatients with T2DM and hypertension visiting hospitals and health centers were included in the study. A proportion to size allocation method was used to determine the required sample size per facility. Data was collected using the 8-item Morisky medication adherence scale. Descriptive statistics and binary logistic regression were used to analyze data. A 95% confidence interval and p≤0.05 statistical significance was considered to determine factors associated with poor medication adherence.

### Results

A total of 409 patients were included in the present study. About 57% of the patients reported that the COVID-19 pandemic has posed negative impacts on either of their follow-up visits, availability of medications, or affordability of prices. And, 21% have reported that they have been affected in all aspects. The overall magnitude of poor medication adherence was 72%. Patients with extreme poverty were more likely to have good medication adherence (AOR: 0.59; 95%C.I: 0.36–0.97), whereas attendance to a health center (AOR: 1.71; 95%C.I: 1.02–2.85), presence of comorbidity (AOR: 2.05; 95%C.I: 1.13–3.71), and current

**Data Availability Statement:** All relevant data are within the paper and its Supporting Information files.

**Funding:** Funding for the study was obtained from Saint Paul's Hospital Millennium Medical College. The funder had no role in study design, data collection and analysis, decision to publish, or preparation of the manuscript.

**Competing interests:** The authors have declared that no competing interests exist.

substance use history (AOR: 11.57; 95%C.I: 1.52–88.05) predicted high odds of poor adherence.

## Conclusion

Over a three-fourth of the patients, in the study setting, have poor adherence to their anti-diabetic and antihypertensive medications. Health facility type, income level, co-morbidity, and current substance use history showed a statistically significant association with poor adherence to medication. Stakeholders should set alternative strategies as perceived impacts of the COVID-19 pandemic on medication adherence are high in the study area.

## Introduction

Adherence to anti-diabetic and antihypertensive medications is an area of interest with marked implications to affect patient management outcomes. The World Health Organization (WHO) defined adherence in 2001 as the extent to which a patient follows medical instructions [1]. Whereas this concept accounts for a broader aspect in the medical practice, its correlation with the management and control of chronic illnesses as diabetes and hypertension is well documented. This is likely because, adherence in this group encompasses multiple dimensions, such as lifestyle changes, medications, patient attitudes, and provider-patient relationships among others [2, 3] with potential interaction in the longer or lifetime frames.

Measuring adherence provides information on patients' behavior to medication and lifestyle practices [4]. Studies have documented that the level of medication adherence in developed nations remained, only, close to 50% [5–7]. This figure is even lower in the developing and middle-income countries owed to the paucity of healthcare inputs and limited access to the services. Reports of adherence to antihypertensive medications from China (65.1%), Gambia (27%), and Seychelles (26%) showed the magnitude and relevance of the problem [8–10].

Medication adherence studies on antidiabetic and antihypertensive medications have been conducted across regions of Ethiopia. Even though the measurement tool and design of the study are similar for most of the studies, there is a considerable variation in terms of the level of adherence by study settings and type of chronic illness considered [11–14]. Reports from the present study setting also revealed that adherence levels to anti-diabetic medications ranged from 51.3% to 76% [15–17].

Currently, when the focus of pharmaceutical companies is on battling the COVID-19 pandemic across the globe, low and middle-income countries (LMICs) like Ethiopia, might be worst affected due to the rather vulnerable and inadequate pharmaceutical manufacturing capacities in these countries to meet their general pharmaceutical needs particularly those for chronic diseases. With limited supply to meet the increased demand created, the market values of medicines for chronic diseases have escalated, making them unaffordable for several patients in LMICs who require them [13]. Besides, adequate therapeutic outcomes of chronic illnesses require a linear adherence to medications [18]. This, in turn, could be linked to accessibility, affordability that can, negatively, be impacted during the outbreak [19].

Also, even though more studies have been conducted on the topic [11–17], potential variation in findings is inevitable due to time, study design, perception, context, the population considered, sample size, and quality of care delivered to patients among others. The purpose of this study is to assess the level of medication adherence, patients' perception of the impact of

the COVID-19 pandemic, and factors associated with poor adherence to anti-diabetic and antihypertensive medications among patients visiting public health facilities in Addis Ababa.

## Methods

### Study setting, design, and period

The study is conducted in Addis Ababa, the capital of Ethiopia. The 2021 estimated population of the city is over 4.8 Million [20]. There exist a total of 12 public hospitals [21] of which six are managed under the federal ministry of health (FMOH) whilst the rest five hospitals and 103 health centers are administered under the Addis Ababa health bureau (AAHB) [22]. The prevalence of diabetes mellitus and hypertension in public health facilities was reported to be 14.8% [23] and 32–34.7% [24, 25] respectively. A cross-sectional study design was conducted from 1st through 30th August 2020 at seven public health facilities to assess the adherence of patients to antidiabetic and antihypertensive medications during the COVID-19 pandemic. One of the facilities (Saint Paul's hospital millennium medical college (SPHMMC)) was a teaching hospital under the ministry of health whereas, one general hospital (Ras Desta Damtew memorial hospital (RDDMH)), and five health centers namely; Arada, Lideta, Nifas Silk Lafto wereda 09, Akaki kality, and Bulbula are administered under the Addis Ababa regional health bureau.

### Populations and inclusion criteria

The source population of this study was; all adult outpatients diagnosed with T2DM and hypertension in Addis Ababa, Ethiopia. All outpatients aged 18 years or above, those who have been on either anti-diabetic or antihypertensive medications for more than six months, visiting the selected public health facilities during the stated study period, and who provided informed consent were eligible in the study. Patients with known or suspected psychiatric problems and those attending to the emergency units were not included.

### Sample size and sampling technique

The minimum sample size was estimated based on reports of adherence to antihypertensive [13] and anti-diabetic medications [15]. Of the two figures (66.7% and 51.3% respectively), the latter produced a higher required sample size. The single population proportion formula with a 95% confidence interval, and 5% tolerable error assumptions was employed in the calculation. Adding a 10% for non-response, a total of 422 participants was required for the study. Hospitals and health centers were selected purposively based on patient flow and socio-demographic diversity of catchment populations. A proportion to size allocation method was used to determine the required number of patients per health facility. Taking into account the COVID-19 risk, participants who consented to take part were recruited and included in the study using a consecutive sampling technique.

### Data collection tool, procedures, and quality

Medication adherence was measured based on the 8-item Morisky medication adherence instrument [26]. Socio-demographic and clinical profiles were included with the questionnaire. After obtaining verbal informed consent, a face-to-face interview, using a pre-tested questionnaire, was done by trained data collectors. An Amharic (an official working language of Ethiopia) translated instrument was used during the data collection and back-translated to English for entry and analysis. Content validity of the tool was ensured by the study team. Supervision was undertaken throughout the data collection period.

## Variables of the study

The dependent variables are adherence level and patients' perception of the COVID-19 impact. On the other hand, the independent variables were socio-demographic variables, such as age, sex, health facility type, education, marital status, income level, current substance use history, presence of close people, number of close people, and Clinical variables, such as presence of comorbidity, disease duration, treatment duration, diagnosis type and presence of sleep disturbance.

## Operational definitions

**Good adherence.**   Refers to a patient's overall score sum of 16 points from the 8-item Morisky's medication adherence scale (MMAS). Items 1 through 7 were coded as 1 = yes, 2 = no except for item 5 in which no was rated as 1 and yes was rated as 2. For item 8, the Likert scaled scores of 1 to 5 were reverse coded as 2 = never and 1 = often to always.

**Poor adherence.**   Refers to the MMAS-8 score summation of 8 through 15 where a patient missing at least one or more of the items of the scale was classified under poor adherence.

**Current history of substance use.**   Patient reporting to have used either one or any combination of alcohol, Khat, or cigarettes in the past three months.

**Close people.**   Refers to family members or relatives whom a patient can rely on or seek for any form of assistance during hard times.

**Comorbidity.**   Was considered when a patient presents with one or more chronic conditions in addition to either diabetes or hypertension.

## Ethical considerations

Ethical approval to conduct this study was obtained from Saint Paul's Hospital Millennium Medical College (SPHMMC) institutional review board (IRB), and the Addis Ababa regional health bureau research ethics review committee. After ethical clearance was sought from the regional health bureau, a support letter was written to the respective health facilities before data collection. Verbal informed consent was obtained from each participant included in the study. Participation in the study was voluntary. Confidentiality of the data obtained was maintained throughout and after completion of the study. No personal identifiers were either included in the tool or were collected during the study.

## Data analysis

After cleaning and coding manually, data was entered and checked for completeness and accuracy in statistical products for a social solution (SPSS) V.26.0. Both descriptive and inferential statistics were applied in the analysis. Tables and figures were used to present descriptive results. A Bi-variable analysis of all potential patient characteristics was done at p≤0.2 for potential association with poor adherence to anti-diabetic and antihypertensive medications. Variables that satisfied the first test were subsequently included in the multivariable logistic regression at a p≤0.05 level of significance. A 95% level of confidence was considered in both cases.

## Results

### Profile of patients

A total of 409 patients were included in the present study. Even though few patients declined to take part during the subsequent recruitment (n = 13), the number of respondents was beyond the minimum requirement. The age of patients ranged from 19 to 95 with a mean of

56.5 and a standard deviation (SD) of 13.4 Years. The majority were in the age group of 45 or above (321,78.5%), females (229,6%), visiting hospitals (224,54.8%), not attended formal education (174, 42.5%), married (259, 63.3%), live with moderate poverty or better (222, 54.3%), have comorbidity (250, 61.0%), with 7 years or less mean duration of disease (277, 67.7%) and 7 years or less mean duration of treatment (257, 62.8%). Most of the participants had no current history of any substance use (92.2%), diagnosed with hypertension only (35.0%), had no sleep disturbance (68.7%), have close people around (88.5%), and live with 3–5 family members (59.7%) (Table 1). Among the comorbidities reported, hypertension (204, 81.6%), diabetes mellitus (179, 71.6%), heart disease (66, 26.4%), hypercholesterolemia (36, 14.4%), chronic asthma (23, 9.2%), and stroke (12, 4.8%) accounted for the top frequencies. A Cronbach's alpha test of the reliability of the scale among the samples also showed an acceptable range for both hospitals ($\alpha = 0.88$) and health centers ($\alpha = 0.63$).

## Patients' perception of the impact of COVID-19 pandemic

About 163(40%) of the patients reported that the COVID-19 pandemic has posed negative impacts on the availability of medications and their follow-up visits, whereas 160(39%) believed that it caused an unaffordable or increased price of medications. Two hundred thirty-four (57%) reported that they have faced one or more of the problems whilst 87(21%) stated that they come across all the three (Fig 1).

## Level of adherence to antidiabetic and antihypertensive medications

The level of adherence to antidiabetic and antihypertensive medications was measured using the 8-item Morisky medication adherence scale. Accordingly, the overall level of adherence was found to be 28% whilst 72% were poorly adherent missing at least one element from the scale (Fig 2).

## Factors associated with poor medication adherence

Lastly, multiple characteristics of patients were tested against the presence of any potential association with poor medication adherence. Patients under extreme poverty were more likely to report a good adherence as compared to patients with moderate poverty or better average monthly income (AOR: 0.59; 95%C.I: 0.36–0.97). On the other hand, patients attending health centers (AOR: 1.71; 95%C.I: 1.02–2.85), having any comorbidity (AOR: 2.05; 95%C.I: 1.13–3.71), and current history of any substance use (AOR: 11.57; 95%C.I: 1.52–88.05) have shown a statistically significant positive association with poor medication adherence (Table 2).

## Discussion

The COVID-19 pandemic has severely affected health systems in general and follow-up service to chronic illnesses in particular [28–30]. Its impact in Sub-Saharan Africa is more pronounced as this region is often characterized by low health system infrastructure coupled with a growing burden of non-communicable diseases [31]. In Ethiopia too, most healthcare services have been disrupted due to the alarming spread of the pandemic and daily loaded terrifying news [32]. Augmented with less attention of the public to regularly practice preventive measures, more patients are still at risk of either its direct or indirect impacts militating their adherence to therapy. It is apparent from the present study that the majority of patients experienced negative impacts on either of their follow-visits, availability, or affordability of medications at least once during the outbreak. It was also noted that about a fifth of the patients in the present sample reported having faced all of the problems.

**Table 1. Profile of T2DM and hypertensive patients visiting chronic care units of public health facilities in Addis Ababa, August 2020.**

| Characteristic | Label | Frequency | % |
|---|---|---|---|
| **Age (yrs.)** | | | |
| | ≤45 | 88 | 21.5 |
| | >45 | 321 | 78.5 |
| **Sex** | | | |
| | Male | 180 | 44 |
| | Female | 229 | 56 |
| **Visiting site** | | | |
| | Health center | 185 | 45.2 |
| | Hospital | 224 | 54.8 |
| **Education** | | | |
| | Not attended formal education | 174 | 42.5 |
| | Grades 1 to 12 | 165 | 40.4 |
| | Diploma or above | 70 | 17.1 |
| **Marital status** | | | |
| | Unmarried | 53 | 13.0 |
| | Married | 259 | 63.3 |
| | Divorced/separated/widowed | 97 | 23.7 |
| **Income level (ETB)** [*] | | | |
| | Extreme poverty | 187 | 45.7 |
| | Moderate poverty or better | 222 | 54.3 |
| **Presence of comorbidity** | | | |
| | No | 159 | 38.9 |
| | Yes | 250 | 61.1 |
| **Disease duration**[**] | | | |
| | ≤7 years | 277 | 67.7 |
| | >7 years | 132 | 32.3 |
| **Treatment duration**[**] | | | |
| | ≤7 years | 257 | 62.8 |
| | >7 years | 152 | 37.2 |
| **Current substance use history** | | | |
| | Yes | 32 | 7.8 |
| | No | 377 | 92.2 |
| **Diagnosis type** | | | |
| | Type 2 diabetes mellitus | 132 | 32.2 |
| | Hypertension | 143 | 35.0 |
| | Both T2DM and hypertension | 134 | 32.8 |
| **Presence of sleep disturbance** | | | |
| | Yes | 128 | 31.3 |
| | No | 281 | 68.7 |
| **Presence of close people** | | | |
| | No | 47 | 11.5 |
| | Yes | 362 | 88.5 |
| **Number of family members** | | | |
| | ≤2 | 79 | 19.3 |
| | 3–5 | 244 | 59.7 |

(*Continued*)

**Table 1.** (Continued)

| Characteristic | Label | Frequency | % |
|---|---|---|---|
| | ≥6 | 86 | 21.0 |

*classified considered based on the World Bank's definition of poverty [27].

** considered based on the mean duration (Yrs.) of diagnosis or initiation of treatment.

Evaluation of adherence to medication level could be regarded as a potential indicator for patients' commitment to reverse an affected state of health [33]. Poor adherence to long-term medications is multifactorial often comprising socio-demographic factors, individual patient-related, therapy-related factors, and the health system among others [34]. As a result, achieving proper adherence remains to be a challenge both in the developed [5, 6] and developing [7–10] countries even though pharmacotherapy remains to be the mainstay of treatment especially among the older population [35].

The present study shows that level of poor adherence, where a patient failed to meet all the recommended criteria, was found to be significantly high (72%). This figure is higher as compared to the level of adherence to antihypertensive medications reported from Southwest Ethiopia (61.8%) by Asgedom and his colleagues [11], in Hawassa (67%) documented by Getnet et al. [12], and in Addis Ababa (66.7%) reported by Tibebu and his colleagues [13]. The highest level of adherence (75%) to antihypertensive medications was reported from Northwest Ethiopia as well [14]. More precisely, the earlier studies have indicated that at least half a proportion of the patients in Addis Ababa had a good level of adherence to their antihypertensive [13] and antidiabetic medications [15–17]. Whereas multiple contributors would result in a poor outcome, the impact of the COVID-19 pandemic six months before the study period was undeniably notable [32, 36]. Most patients have also implied in their report of perceived negative influence the outbreak had posed in terms of meeting with follow-up appointments, accessing medicines, and affording for prices. These factors are mentioned to account for a remarkable role in patients' poor adherence to therapy [37, 38].

Studies documented that various factors have a link with poor adherence to medications among patients with chronic illness. These may include; lack of involvement of family and friends [39], economic difficulties [40], poor relationship between professional and patients [41], and side effects as well as the complexity of regimens [41, 42]. Furthermore, while patients with asymptomatic diseases have less incentive to adhere to medications, the presence of multiple comorbid conditions that are treated with more drugs can also impair attaining

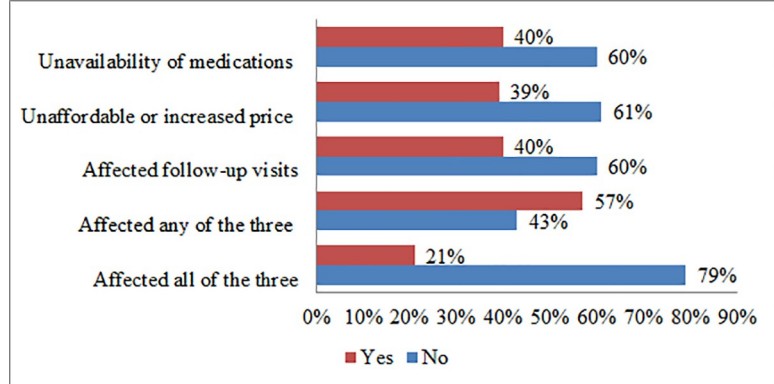

**Fig 1. Perception of T2DM and hypertensive patients on the impact of the COVID-19 pandemic visiting chronic care units of public facilities in Addis Ababa, August 2020.**

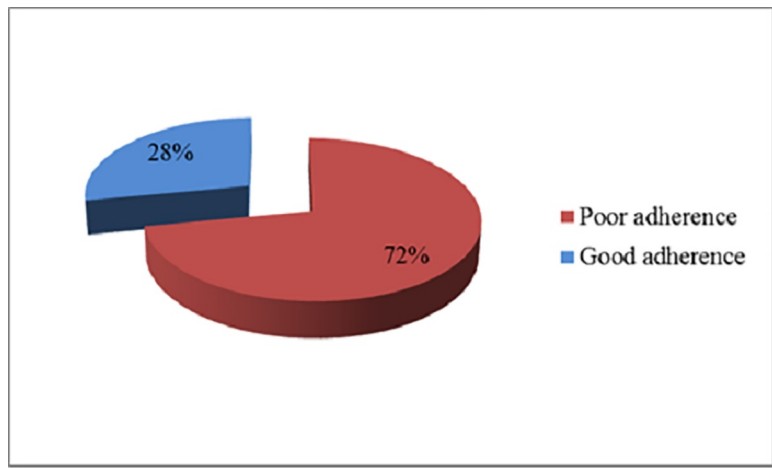

**Fig 2. Adherence level to antidiabetic and antihypertensive medications among patients visiting public health facilities during the COVID-19 pandemic in Addis Ababa, Ethiopia, August 2020.**

**Table 2. Factors associated with poor adherence to antidiabetic and antihypertensive medications among patients visiting public health facilities during the COVID-19 pandemic in Addis Ababa, Ethiopia, August 2020.**

| Variable | Label | Level of adherence | | COR (95%CI) | AOR (95%CI) |
|---|---|---|---|---|---|
| | | Good (n) | Poor (n) | | |
| **Facility level** | | | | | |
| | Health centers | 38 | 147 | 1.98(1.27–3.12) | 1.71(1.02–2.85)* |
| | Hospitals | 76 | 148 | 1 | 1 |
| **Education** | | | | | |
| | Not attended formal education | 35 | 139 | 1.94(1.04–3.62) | 1.85(0.94–3.63) |
| | Primary/secondary education | 56 | 109 | 0.95(0.53–1.73) | 0.96(0.50–1.84) |
| | College/University education | 23 | 47 | 1 | 1 |
| **Average monthly income (ETB) [d]** | | | | | |
| | Extreme poverty | 58 | 129 | 0.75(0.49–1.16) | 0.50(0.27–0.89)* |
| | Moderate poverty or better | 56 | 166 | 1 | 1 |
| **Comorbid condition** | | | | | |
| | No | 56 | 103 | 1 | 1 |
| | Yes | 58 | 192 | 0.56(0.36–0.86) | 2.05(1.13–3.71)* |
| **Current history of substance use** | | | | | |
| | Yes | 1 | 31 | 13.27(1.79–98.39) | 11.44(1.50–87.11)* |
| | No | 113 | 264 | 1 | 1 |
| **Diagnosis type** | | | | | |
| | T2DM | 42 | 90 | 0.59(0.34–1.03) | 1.06(0.52–2.16) |
| | Hypertension | 43 | 100 | 0.64(0.37–1.11) | 0.92(0.48–1.77) |
| | Both T2DM and hypertension | 29 | 105 | 1 | 1 |
| **Presence of sleep disturbance** | | | | | |
| | Yes | 28 | 100 | 1.58(0.97–2.57) | 1.44(0.86–2.42) |
| | No | 86 | 195 | 1 | 1 |
| **Presence of close people around** | | | | | |
| | No | 7 | 40 | 2.40(1.04–5.52) | 2.03(0.84–4.90) |
| | Yes | 107 | 255 | 1 | 1 |

*indicates statistical significance at P≤0.05. COR: crude odds ratio; AOR: adjusted odds ratio.

proper adherence [34, 43, 44]. The earlier studies done in Ethiopia have indicated that such contributors, as comorbidity [11–13], age group [12–14], level of knowledge about disease and medication [12–14, 16], and level of education [15, 16] were reported to be among the factors associated with medication adherence.

In paradox with the popular belief [40, 45], lower-income was found to be associated with higher odds of good adherence to anti-diabetic and antihypertensive medications in the current setting. Though the connection of income and health is well established in terms of either direct effect on material fulfillment or indirectly through ensuring social participation [46], the present association could be confounded with non-income factors. In the current Ethiopian context, all patients under the poverty line (45.7% of the respondents) are waived insurance fees that may improve their service utilization rate and access to medicines. Yet, adherence is a behavior of composite factors [34] that can be altered even apart from gaining access to service and medications. Taking into account non-compliance consequences, poor patients might tend to apply medical advice or maintain good relations with providers.

On the other hand, attending a health center, having any comorbid condition and current history of any substance use have shown a statistically significant positive association with poor medication adherence. An increase in the odds of poor adherence among patients attending health centers would be related to patients' health-seeking behavior, the severity of complications, patient loads, or perceived poor availability of COVID-19 preventive measures at these settings. That could be likely because, as primary healthcare units in the Ethiopian health system, health centers serve to catchment populations that do not need advanced diagnosis and treatment. This is in line with other studies that reported that having a comorbid condition [11] and substance use history [11, 47] were associated with high odds of poor medication adherence. This could also be attributed to the fact that patients may either not be able to comprehend the outcome of their disease and benefit of adherence to medications [48], experience side effects of polypharmacy [49] or fail to practice a healthy lifestyle.

This study has tried to present a city-wide comprehensive report on the medication adherence practice of patients visiting public health facilities in Addis Ababa, Ethiopia. Apart from being able to include various strata of facilities and patients, the findings can be easily compared to previous figures to evaluate the impact imposed by the COVID-19 pandemic on medication adherence. However, the observed association between dependent and independent variables may suffer from temporality. Inclusion of the only patients who were available during the study period could also pose a selection bias that might have affected the study results.

## Conclusion

A significant proportion of patients with T2DM and hypertension in Addis Ababa have experienced negative impacts on either of their follow-visits, availability, or affordability of medications at least once during the COVID-19 outbreak. Over a three-fourth of the patients had poor adherence to their medications. Facility type, average monthly income, level of education, presence of comorbidity, and current histories of any substance use have shown a statistically significant association with poor adherence to antidiabetic and antihypertensive medications.

All concerned health authorities should take into account, and set multidisciplinary strategies to prevent impacts of the COVID-19 pandemic on medication adherence of patients with chronic illnesses.

## Supporting information

**S1 File.**
(DOCX)

**S2 File.**
(DOCX)

## Acknowledgments

The authors would like to thank all patients who provided verbal consent to participate in this study. We also appreciate the kind assistance and facilitation gotten from all health facilities during the data collection process.

## Author Contributions

**Conceptualization:** Tariku Shimels.

**Data curation:** Rodas Asrat Kassu, Gelila Bogale.

**Formal analysis:** Melsew Getnet.

**Funding acquisition:** Tariku Shimels.

**Investigation:** Gelila Bogale, Mahteme Bekele, Mebratu Abraha.

**Methodology:** Tariku Shimels, Mahteme Bekele, Melsew Getnet, Abrham Getachew, Zewdneh Shewamene, Mebratu Abraha.

**Project administration:** Tariku Shimels, Rodas Asrat Kassu, Gelila Bogale, Mahteme Bekele, Mebratu Abraha.

**Supervision:** Tariku Shimels, Rodas Asrat Kassu, Mahteme Bekele, Melsew Getnet, Abrham Getachew, Zewdneh Shewamene.

**Validation:** Tariku Shimels.

**Writing – original draft:** Tariku Shimels, Rodas Asrat Kassu, Zewdneh Shewamene.

**Writing – review & editing:** Tariku Shimels, Rodas Asrat Kassu, Gelila Bogale, Melsew Getnet, Abrham Getachew, Mebratu Abraha.

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
