## [Decision Letter · Decision Letter 0]

21 Jan 2021

PONE-D-20-34174

Magnitude and associated factors to medication poor adherence among diabetic and hypertensive patients visiting public health facilities in Ethiopia during the COVID-19 pandemic

PLOS ONE

Dear Dr. Shimels,

Thank you for submitting your manuscript to PLOS ONE. After careful consideration, we feel that it has merit but does not fully meet PLOS ONE’s publication criteria as it currently stands. Therefore, we invite you to submit a revised version of the manuscript that addresses the points raised during the review process.

Please note that the comments by the reviewers are quite extensive. Please make sure that you adequately address them. Some parts of the introduction and the discussion could be shortened by having them much more concise. This will allow to extend on several aspects as required by the reviewers.

We look forward to receiving your revised manuscript.

Kind regards,

Hans-Peter Brunner-La Rocca, M.D.

Academic Editor

PLOS ONE

Journal Requirements:

Furthermore, within the Methods section, please provide a description of how participants were recruited for the study.

4. Please include in your Methods section (or in Supplementary Information files) the participating hospitals/institutions. Furthermore, please provide additional clarifications as to whether IRB of all participating institutions approved the study prior to data collection.

5. Please ensure that you include a title page within your main document. We do appreciate that you have a title page document uploaded as a separate file, however, as per our author guidelines (http://journals.plos.org/plosone/s/submission-guidelines#loc-title-page) we do require this to be part of the manuscript file itself and not uploaded separately.

Reviewers' comments:

Reviewer's Responses to Questions

**Comments to the Author**

1. Is the manuscript technically sound, and do the data support the conclusions?

Reviewer #1: Yes

Reviewer #2: Partly

2. Has the statistical analysis been performed appropriately and rigorously? 

Reviewer #1: I Don't Know

Reviewer #2: Yes

3. Have the authors made all data underlying the findings in their manuscript fully available?

Reviewer #1: No

Reviewer #2: Yes

4. Is the manuscript presented in an intelligible fashion and written in standard English?

Reviewer #1: No

Reviewer #2: No

5. Review Comments to the Author

Reviewer #1: The authors present an original reasearch article entiteled: Magnitude and associated factors to medication poor adherence among diabetic and hypertensive patients visiting public health facilities in Ethiopia during the COVID-19 pandemic.

The manuscripts reads like a first version manuscript. It needs to be rewritten with clearer senteces. A native speaker should edit the manuscript.

Abstract: the abstract is well written but quite long. Please be consise.

Introduction: THe introduction reads as a textbook with little conffection between the paragrafs.

The first paragraph starts with defining the WHO criteria for adherence. What does adherence refer to. Such a paragfar is not appealing to read further. I suggest to start with why drug adherence for hyperntension and diabetes is of interenst to study.

The paragraphs discussing the literature may be more suitable for the discussion (paragraf 3 and 4). Could be in 2 or 3 sentecences in the introduction.

Methods:

"A cross sectional study design was conducted in August 2020 among seven public health facilities (two hospitals and five health centers) to assess adherence of patients to antidiabetic and antihypertensive medications during the COVID-19

pandemic. " Was it from the start of august till the end of august? Wat was the time window for the study? Who assessed the data?

Wat is there an approval from the Medical ethical commitee?

Where only chronic patients included or also patients with hypertension and DM de novo.

"Content validity was of the tool was ensured by study team. Cronbach’s

alpha test of reliability of the scale among pretested samples showed acceptable range for both

hospitals (α=0.88) and health centers [α=0.63]. " This is describtion of the results. I suggest to transfer it to the results section.

The descrioption of the adherence is unclear for readers who are not familiar with the morskey score. Perhaps a table would be more suitable.

Results:

Please put tables at the end of the paper.

The description of the results is not conssitent. For example: AOR: 0.58; 95%C.I: 0.35-0.97 and AOR: 0.50; 95% C.I:0.27-0.89

Discussion:

Can be extended with the literature discussion from the introduction.

Please start the discussion with a short discription of the results.

The conclusion is again a summary of the results, please make it a conclusion and add a clear recommendation paragraph.

Please add in the discussion a clear strenght and limitation paragraph.

Reviewer #2: This study investigated the adherence to medication in patients with diabetes and hypertension in Ethiopia during the COVID-pandemia. Only app. ¼ of the patients were fully adherent. COVID was mentioned in app. 40% of the patients as causing some reduction in adherence. Other factors could be identified to increase or reduce adherence.

Some comments:

Introduction:

good summary. However, the authors could highlight a bit more what the added value of their work is, in addition to investigating the situation during the COVID pandemia, as compared to the papers cited in the introduction (e.g. differences in the setting as I can imaging that the populations seen in different settings differ, or why numbers of other studies may be not representative, or that the population of this study is the same as that of one of the other studies, allowing to investigate the direct impact of COVID on adherence).

Methods:

Where all patients including during the time period or was it a selected group of patients? Did all patients consent to participate in the study? Probably not, To what extend did those not included differ from those that participated in this study? It may be helpful to have a figure explaining the patient flow.

The sample size calculation is unclear. What was the aim / target for the calculation? Please be more precise. I do not understand it at present.

It would be helpful to add the questionnaire, at least as supplementary file. This makes it also easier to understand exactly how good adherence was defined.

Is it correct that participating patients did not give written informed consent? If yes, was this an exemption specifically approved by the Ethical Committee a priori?

Were all variables included in multivariable analysis or were there any selection criteria? If no selection criteria applied, the power may be not sufficient for multivariable analysis given the number of independent variables. In this case, I would recommend to first perform a univariable analysis and only include those with a e.g. P<0.1 in the multivariable analysis. At present, it is not clear why some variables are mentioned in table 2 and others not.

Results:

How was the presence of close people defined? How was co-morbidity defined? Would it not be better to name the prevalence of co-morbidities (at least for the most prevalent ones)? In addition, it would be interesting with which medication patients were treated.

Figure 1. % instead of absolute numbers would be better.

As far as I understand, an OR of <1.0 indicates less non-adherence, i.e. better adherence. I think that it is easier to understand if you only better adherence (with an OR > 1) and worse adherence (with an OR <1). This is easier to understand. This obviously also applies to the according parts of the discussion (e.g. association with better adherence instead of association with lower odds of poor adherence).

Would it not be better to use the poverty line as cut-off instead of the median within the study, particularly as this is also related to the discussion?

Please define AOR and COR in table 2.

Discussion:

It is a bit confusing that you refer that 40% are negatively affected by the pandemia regarding non-adherence, but in fact, there were 57%. This applies also to the abstract. This is confusing. In addition, you do not need to repeat results in the discussion (use e.g. the majority instead).

Please discuss your findings in the light of previous findings in your country. You only do so regarding the overall adherence. It would be interesting to hear if factors associated with better adherence were similar in other studies and what potential explanations for the differences are.

The finding and discussion of low income being associated with better adherence is interesting. What do other studies say on this and what may be the differences with your findings? This is missing. The findings related to co-morbidites etc. are in line with your findings. This needs to be mention and is separate to the discussion about impact of low income.

Have a separate paragraph regarding limitations. You also need to add that the number of included patients is limited and other important factors may have been missed because of this limitation. In addition, the selection of patients may be a limitation (not only that inclusion was limited to one month), but this is difficult to judge as nothing is reported in this regard (see above).

Minor comments:

There are quite some spelling and grammatical mistakes (e.g. incorrect use of tenses). Please check and correct.

6. PLOS authors have the option to publish the peer review history of their article (what does this mean?). If published, this will include your full peer review and any attached files.

Reviewer #1: No

Reviewer #2: No

---

## [Decision Letter · Decision Letter 1]

25 Feb 2021

PONE-D-20-34174R1

Magnitude and associated factors of poor medication adherence among diabetic and hypertensive patients visiting public health facilities in Ethiopia during the COVID-19 pandemic

PLOS ONE

Dear Dr. Shimels,

Thank you for submitting your manuscript to PLOS ONE. After careful consideration, we feel that it has merit but does not fully meet PLOS ONE’s publication criteria as it currently stands. Therefore, we invite you to submit a revised version of the manuscript that addresses the points raised during the review process.

See comments by reviewer #2. These are not major issues, but you still should address them. In particular, please provide the adequate information about the power calculation and have your manuscript copy-edited by a native speaking person.

We look forward to receiving your revised manuscript.

Kind regards,

Hans-Peter Brunner-La Rocca, M.D.

Academic Editor

PLOS ONE

Journal Requirements:

Reviewers' comments:

Reviewer's Responses to Questions

**Comments to the Author**

1. If the authors have adequately addressed your comments raised in a previous round of review and you feel that this manuscript is now acceptable for publication, you may indicate that here to bypass the “Comments to the Author” section, enter your conflict of interest statement in the “Confidential to Editor” section, and submit your "Accept" recommendation.

Reviewer #1: All comments have been addressed

Reviewer #2: (No Response)

2. Is the manuscript technically sound, and do the data support the conclusions?

Reviewer #1: Yes

Reviewer #2: Yes

3. Has the statistical analysis been performed appropriately and rigorously? 

Reviewer #1: Yes

Reviewer #2: Yes

4. Have the authors made all data underlying the findings in their manuscript fully available?

Reviewer #1: Yes

Reviewer #2: Yes

5. Is the manuscript presented in an intelligible fashion and written in standard English?

Reviewer #1: Yes

Reviewer #2: No

6. Review Comments to the Author

Reviewer #1: The authors adressed all comments adequately.

The respons en manuscript can be improved to a more neat version, but probably with the editing phase this matter can be adressed.

Reviewer #2: I would like to thank the authors for the improvement in their manuscript. Most issues are resolved sufficiently, but there are still some remaining.

Thus, the authors still fail to provide sufficient information regarding the power calculation. The authors provide references from previous studies, but they fail to exactly mention their assumptions that led to the required sample size.

The use of the English language has improved, but the manuscript still contains mistakes. As there is no copy-editing for PLOS ONE, the manuscript must be provided in good English without mistakes. I understand that this may be difficult for the authors. Therefore, they need to have the manuscript corrected by a person that is (almost) native English speaking.

Please provide the information that written informed consent was not required.

According to the instruction to authors, tables should be placed directly after the paragraph in which they are cited (the recommendation by the other reviewer was not correct).

7. PLOS authors have the option to publish the peer review history of their article (what does this mean?). If published, this will include your full peer review and any attached files.

Reviewer #1: No

Reviewer #2: No

---

## [Author Response · Author response to Decision Letter 1]

11 Mar 2021

Thank you for the email, and requesting us to make a second revision of the manuscript titled, “Magnitude and associated factors to medication poor adherence among diabetic and hypertensive patients visiting Public health facilities in Ethiopia during the Coronavirus pandemic” submitted for potential publication in your journal. We appreciate and thank the reviewers for the comments raised. The following changes were made to the document.

i) Explanation to the sample size calculation: two earlier reported figures on the level of adherence to anti-DM and antihypertensive medications were used for the calculation. Though the two reported figures were tested for, the sample size which was obtained based on adherence to anti-DM (i.e, adherence to anti-DM, p=51.3%) was higher (384) and considered as the minimum required sample size. As a single population observational [cross-sectional] study, the confidence interval method was considered to compute the minimum sample size. It is also apparent that 95% C.I [which means a 5% allowed type I error], 5% level of precision, and a Point estimate for level of adherence to anti-DM and anti-hypertensive medications (reported earlier) was included reasonably. This has been presented in the methods section as:

 The minimum sample size was estimated based on reports of adherence to antihypertensive [13] and anti-diabetic medications [15]. Of the two figures (66.7% and 51.3% respectively), the latter produced a higher required sample size. The single population proportion formula with a 95% confidence interval, and 5% tolerable error assumption was employed in the calculation. Adding a 10% for non-response, a total of 422 participants was required for the study………… 

ii) Regarding language issue, and editorial mistakes

The authors and an additional academician (out of the SPHMMC staff) have revised the grammar as well as typological errors thoroughly, and have fixed some issues. Unfortunately, we were unable to find one who will be both a native speaker and understands the subject area. We hope that the editorial quality is quite improved as it presents now, but still willing to receive further comments. All details are presented in a ‘red font’ highlight within the ‘manuscript with track change’ version. 

We appreciate for the time and consideration to our work, and look forward to hearing from your end. 

Tariku Shimels, corresponding author

---

## [Editor Report · Decision Letter 2]

15 Mar 2021

Magnitude and associated factors of poor medication adherence among diabetic and hypertensive patients visiting public health facilities in Ethiopia during the COVID-19 pandemic

PONE-D-20-34174R2

Dear Dr. Shimels,

We’re pleased to inform you that your manuscript has been judged scientifically suitable for publication and will be formally accepted for publication once it meets all outstanding technical requirements.

Kind regards,

Hans-Peter Brunner-La Rocca, M.D.

Academic Editor

PLOS ONE
---

## [Editor Report · Acceptance letter]

29 Mar 2021

PONE-D-20-34174R2 

Magnitude and associated factors of poor medication adherence among diabetic and hypertensive patients visiting public health facilities in Ethiopia during the COVID-19 pandemic 

Dear Dr. Shimels:

I'm pleased to inform you that your manuscript has been deemed suitable for publication in PLOS ONE. Congratulations! Your manuscript is now with our production department. 

Kind regards, 

on behalf of

Dr. Hans-Peter Brunner-La Rocca 

Academic Editor

PLOS ONE